# Assessing Awareness of Colorectal Cancer Symptoms among Outpatients: A Cross-Sectional Study at a Hospital in Vietnam

**DOI:** 10.3390/healthcare11233063

**Published:** 2023-11-29

**Authors:** Chon Kim Nguyen, Hieu Minh Phan, Chao-Hsien Lee, Lan Anh Thi Do

**Affiliations:** 1Department of Day Service Unit, Hoa Hao Medic Company Limited, Ho Chi Minh City 700000, Vietnamhieuphanminh76@gmail.com (H.M.P.); 2Department of Nursing, Meiho University, Pingtung 912009, Taiwan; 3Department of Epidemiology, Faculty of Public Health, Pham Ngoc Thach University of Medicine, Ho Chi Minh City 700000, Vietnam; lananhytcc08@gmail.com

**Keywords:** colorectal cancer (CRC), awareness, risk factors, bowel CAM, outpatients, Vietnam

## Abstract

Colorectal cancer (CRC) is a prevalent cancer globally, including in Vietnam where its incidence is rapidly increasing. The aim of this study was to evaluate the awareness of signs, symptoms, and risk factors of colorectal cancer among outpatients at Hoa Hao Medic Company Limited, Ho Chi Minh City, Vietnam. A cross-sectional study was conducted, and a total of 441 people who visited Hoa Hao Medic Company Limited for a general health check-up and voluntarily agreed to participate in this study were recruited through the convenience sampling method. Data were collected through face-to-face structured interviews using the Bowel CAM questionnaire. According to the results, the highest percentage of well-recalled warning signs were “change in bowel habit” (36.7%) followed by abdominal pain (35.4%). Other warning signs such as weight loss and rectal bleeding were reported by 19.0% and 18.1% of participants, respectively. Over 42% of participants stated that they were unaware of any signs or symptoms. The most commonly identified risk factors for CRC were pollution (66.9%), genetics (50.6%), and an unhealthy/poor diet (53.7%) for unprompted questions. The overall awareness score of participants was 3.46/9 (SD ± 2.91) for signs and symptoms of CRC and 5.90/10 (SD ± 1.62) for risk factors. Univariate linear regression identified education level and occupation as predictors of higher CRC awareness. In conclusion, this study highlights that overall awareness of CRC is low among outpatients at Hoa Hao Medic Company Limited. Strategies to increase awareness, knowledge, and education programs are necessary to promote early detection of CRC and reduce its burden in Vietnam.

## 1. Introduction

Colorectal cancer (CRC) is a widespread cancer that ranks fourth among the most common cancers and as the third leading cause of cancer-related deaths globally. The burden of CRC continues to rise, with almost 2 million new cases and 1 million deaths recorded yearly [1]. In the United States, an estimated 50,260 people died from CRC in 2017, with a slightly higher death rate in men than women [2]. By 2030, the global burden of CRC is expected to increase by 60%, leading to more than 2.2 million new cases and 1.1 million deaths [3].

Colorectal cancer represents a significant global health concern, with its incidence and mortality rates exhibiting notable variations across different regions. In the worldwide context, research analyzed data from various countries and revealed that countries with higher Human Development Index scores, such as the Republic of Korea, Slovakia, and Hungary, had elevated colorectal cancer incidence rates [4]. This trend was similarly observed in the Asian-focused study, suggesting a consistency in the relationship between socio-economic development and colorectal cancer incidence [5]. These findings emphasize the importance of socio-economic factors, including Human Development Index, in understanding the prevalence of colorectal cancer, both worldwide and in the Asian region [4,5].

CRC is often considered a lifestyle-related disease that is increasingly prevalent in developing countries that are adopting western lifestyles. Risk factors for CRC include a sedentary lifestyle, obesity, tobacco and alcohol consumption, and red meat consumption [6].

In Vietnam, CRC is also becoming more prevalent, with a significant rise in cases in recent years [1]. Age, personal history of inflammatory bowel disease, cancer, and family history of CRC, including genetic conditions such as polyps and CRC, are all associated with an increased risk of developing the disease [7].

There is a significant difference in survival rates depending on the stage of colorectal cancer at diagnosis. The survival rate after 5 years was reported to be as high as 90% if the cancer is diagnosed at an early stage [8]. Unfortunately, a report from the Gulf Center for Cancer Control and Prevention (2011) showed that 60% of colorectal cancer cases were diagnosed at a late stage [9]. In the UK, the average 5-year survival time for colorectal cancer was 51%, but the survival rate varied with the stage of disease upon diagnosis, with over 93% of patients surviving a localized diagnosis (Dukes stage segment A) compared with only 7% of them with distant metastases [8]. In Taiwan, the highest 5-year survival rate was 69.11% in colon cancer patients, followed by 68.66% in colorectal cancer patients and 67.90% in rectal cancer patients [10]. Currently, in the UK, only 13% of cases are diagnosed at an early stage [11]. To reduce this difference, it is crucial to have knowledge of the disease for early diagnosis, which would improve patient survival [6].

Previous studies have shown that a lack of knowledge of primary symptoms and risk factors of CRC affects participation in screening and leads to late diagnosis of the disease [12,13,14]. Thus, increasing awareness of the disease and its risk factors may increase participation in screening programs and reduce the economic and social burden of CRC [6].

The aim of this study is to assess the awareness and related factors about symptoms and risk factors of colorectal cancer among outpatients of Hoa Hao Medic Company Limited.

## 2. Methodology

### 2.1. Study Design, Setting, and Sampling

A cross-sectional study was conducted, involving a convenience sample of 441 participants who attended the Department of Day Service Unit at Hoa Hao Medic Co., Ltd. (Ho Chi Minh City, Vietnam) Between August and November 2021. Inclusion criteria for this study encompassed all Vietnamese individuals who availed themselves of the Department of Day Service Unit’s services for general health check-ups during the data collection period and willingly agreed to participate. Exclusion criteria for this study were patients with a prior history of colorectal cancer or those facing mental and communication issues that impeded their ability to respond to information collection forms.

To mitigate potential biases inherent to cross-sectional studies, all participants meeting the inclusion criteria underwent face-to-face interviews using the Bowel CAM questionnaire. These interviews were conducted by trained interviewers during the participants’ presence in the Day Service Unit.

### 2.2. Data Collection Tools

Participants who met the admission criteria were approached and provided with research information. If the participants agreed, they were invited to sign the consent form to participate in this study. Researchers conducted face-to-face interviews using the Bowel Cancer Awareness Measure (CAM) toolkit, version 2.1, which was developed by the University College London and Cancer Research UK in 2011, and the validity was assessed by 16 bowel cancer specialists and 35 university experts. The toolkit demonstrated good internal reliability, with Cronbach’s alpha exceeding 0.7 for all components [15]. The questionnaire was translated into Vietnamese and divided into two sections. Section A consisted of seven questions investigating the demographic characteristics of the participants, including age, gender, ethnicity, marital status, level of education, job, and family history of colorectal cancer. Section B consisted of five main questions that focused on the participant’s knowledge of the early symptoms and risk factors of colorectal cancer. The total scores, calculated from a combination of 19 items, ranged from 0 to 19. It consisted of 9 items for signs and symptoms and 10 items for risk factors, scored as 1 point for a “correct” response, 0 points for an “incorrect” response, and 0 points for an “unclear” response [16,17].

### 2.3. Ethical Considerations

The data collection was conducted after obtaining approval from the IRB Biomedical Research Committee, Hoa Hao Medic Company Limited, Ho Chi Minh City, Vietnam, with IRB No. 02/2021/HDDD-YTHH.

### 2.4. Statistical Analysis

Data were analyzed using IBM (SPSS) Statistics 26.0 software. Descriptive statistics including frequency, percentage, mean, and standard deviation were used to describe participants’ demographic characteristics and the level of awareness. Differences in the mean awareness scores among different subgroups were compared using two statistical methods: the independent *t*-test and ANOVA with Scheffé’s method. The independent *t*-test was used to compare mean scores between two specific subgroups, while ANOVA with Scheffé’s method was employed to assess mean differences among three or more subgroups. Additionally, univariate linear regression was used to identify predictors of colorectal cancer among outpatients. Statistical significance was set at *p* < 0.05.

## 3. Results

Four hundred and forty-one (441) adults between the ages of 18 and 88 participated in the present study. Table 1 describes the socio-demographic characteristics and history of cancer of the total 441 outpatients. The majority of the participants were of Kinh ethnicity (99.1%), with a mean age of 46.41 years (SD ± 13.45). The gender distribution of the participants was approximately equal, with slightly more females (57.4%). The majority of participants were married (82.8%) and employed full-time (33.6%), with a high school education level (35.1%). Additionally, 5.2% of respondents had a personal history of cancer, and 12% reported a family history of cancer.

Table 2 presents the results of the recall of symptoms and risk factors for CRC. The majority of participants were unable to recall all warning signs and symptoms of CRC. The mean score of unprompted recall from participants was low, with less than two signs or symptoms per participant (mean: 1.9; SD ± 1.18). The most well-remembered warning sign was a change in bowel habit, mentioned by 36.7% of participants, followed by abdominal pain with 35.4%. Weight loss and back passage bleeding were recalled by 19.0% and 18.1% of participants, respectively, and more than 42% reported not knowing any signs or symptoms. In terms of identifying risk factors for CRC, the average recall was more than three risk factors (mean: 3.29; SD ± 1.48). The most commonly identified risk factors were pollution (66.9%), genes (50.6%), and unhealthy/poor diet (53.7%). Respondents also named lifestyle, drinking alcohol, smoking, and not doing enough exercise as risk factors, with percentages of 41.7%, 39.7%, 19.0%, and 13.8%, respectively. However, 6.1% reported not knowing any risk factors.

Table 3 shows the total mean score of awareness about CRC among outpatients, including mean score of awareness about signs and symptoms (Q2), mean score of awareness about risk factors of CRC (Q6), and overall mean score of awareness about CRC (Q2 + Q6). We calculated a score only for a participant’s correct response to each close-ended (prompted) question, those being question two and question six. Total scores ranged from 0 to 19 (the maximum score of signs and symptoms was 9 and risk factors was 10). According to the results, the mean score of awareness about signs and symptoms was 3.46 (±2.91), the mean score of awareness about risk factors was 5.90 (±1.62), and the overall mean score of awareness about symptoms and risk factors of CRC was 9.36 (±4.53).

Table 4 presents the association between patients’ demographic characteristics and awareness of colorectal cancer signs, symptoms, and risk factors. The results indicated no significant association between awareness of colorectal cancer signs, symptoms, and risk factors and demographic factors, including gender, age group, ethnicity, marital status, and personal history of cancer (personal cancer, partner cancer, close family’s cancer, relatives’ cancer).

Factors that were found to associate with awareness of colorectal cancer symptoms were level of education (*p* = 0.002), history of cancer (friend’s cancer) (*p* = 0.013), and kind of job (*p* = 0.015). Regarding the history of cancer, patients having friends with cancer exhibited a higher mean score of awareness about colorectal cancer symptoms (5.18, 95% CI: 4.00–6.36). For the level of education, patients with higher education demonstrated higher mean scores of awareness about colorectal cancer, with the highest scores observed among patients graduating from university and postgraduate programs, and the lowest among patients with a primary school education. Scheffé’s method indicated that patients graduating from university had a significantly higher mean score of awareness compared to those with only an elementary school education. As for kind of job, retired patients had the highest mean score of awareness about colorectal cancer symptoms (4.34, 95% CI: 3.45–5.23), while patients studying had the lowest mean score of awareness about symptoms (2.30; 95% CI: 0.51–4.09) and risk factors (5.30, 95% CI: 3.99–6.61). Employed full-time and retired patients had the highest mean score of awareness about colorectal cancer risk factors (6.24, 95% CI: 5.98–6.51).

Factors associated with awareness of colorectal cancer risk factors were level of education (*p* < 0.001) and kind of job (*p* = 0.003). For the level of education, patients with higher education demonstrated higher mean scores of awareness about colorectal cancer. Scheffé’s method indicated that patients graduating from university had a significantly higher mean score of colorectal cancer awareness compared to patients who graduated from high school, secondary school, or elementary school. Moreover, patients with postgraduate degrees showed higher mean scores of colorectal cancer awareness compared to patients graduating from elementary school or secondary school. As for kinds of job, the ANOVA test showed a significantly different mean score of colorectal cancer awareness among kinds of job (*p* = 0.003). However, the Scheffé test did not reveal any significant differences among kinds of job.

Table 5 presents the results of univariate linear regression analysis, examining the predictors of overall awareness of colorectal cancer among patients. The results indicate that level of education and types of employment (part-time vs. full-time) are significantly associated with overall awareness of colorectal cancer among patients. Patients with a postgraduate degree (mean: 11.16 ± 3.72), university degree (mean: 11.02 ± 3.94), or high school diploma (mean: 9.1 ± 3.35) had significantly higher mean scores of awareness of CRC than patients who graduated from elementary school (mean: 7.78 ± 3.51), with *p* < 0.05. In terms of employment status, compared to full-time employed patients (mean: 9.76 ± 4.18), part-time employed patients (8.38 ± 3.38) had a statistically significant lower overall awareness, with a mean difference of −1.38 (95% CI: −2.48 to −0.27), *p*-value = 0.015. However, there was no statistically significant difference in overall awareness with respect to patients’ demographic characteristics such as gender, age group, ethnicity, marital status, or cancer status of self, parents, relatives, and friends.

## 4. Discussion

In our study, the most well-recalled warning signs of colorectal cancer among participants were change in bowel habit, abdominal pain, weight loss, and back passage bleeding. These findings were consistent with the results of other studies conducted [18,19,20]. However, over 42% of the participants in our study reported that they did not know any signs or symptoms of colorectal cancer. These results were higher than those reported in a study by McCaffery et al. (2003), in which 24% of respondents were unable to identify any warning signs of colorectal cancer [21]. Another study by Chong et al. (2015) found that 54% of participants could not name any signs or symptoms of CRC [22].

The most commonly identified risk factor for colorectal cancer that cannot be modified was genes, with 50.6% of participants choosing this option. This may be because study participants understand that cancer can be hereditary [23]. Another study found that individuals with a first-degree relative (parent, sibling, or child) who has been diagnosed with CRC have two to four times the risk of developing the disease compared to people without this family history, depending on the age at diagnosis and number of affected relatives [24]. However, very few participants in our study chose other risk factors that cannot be modified, such as ethnicity, age, and sex. This is an aspect that needs attention in health education and awareness raising for the population, as the risk of colorectal cancer increases with age, and incidence rates are approximately 30% higher in men than in women, while mortality rates are approximately 40% higher [2]. It is important for the public to be aware of these risk factors so that they can take appropriate measures for the early detection and prevention of colorectal cancer.

Modifiable factors are particularly important because more than half (55%) of colorectal cancers in the US are attributable to potentially modifiable risk factors [25]. In our study, the majority of participants answered pollution as the highest risk factor with 66.9%. This is also consistent with research by Do et al. (2015), where 64.3% of respondents chose pollution as a risk factor [26]. However, knowledge of other modifiable risk factors such as eating red or processed meat, not getting enough fiber, and being overweight was particularly poor in our study, consistent with previous findings [27,28]. This demonstrates that there is still a long way to go in educating the public about the association between living a healthy lifestyle and cancer risk. It is estimated that more than a quarter (27%) of colorectal cancer cases could be prevented through increasing fiber intake and reducing the consumption of red or processed meat. In addition, almost one-seventh (14%) and one-fifth (12%) of possible colorectal cancer cases could be avoided through proper management of excess weight [29,30,31].

The mean score of awareness about symptoms and risk factors of CRC in this study was low, with an overall mean score of 9.36 (±4.53). The mean score for awareness about signs and symptoms was particularly low at 3.46 (±2.91), while the mean score for awareness about risk factors was 5.90 (±1.62). These findings are consistent with other studies conducted among at-risk populations in the Middle East region, such as in Qatar [16]. In Bahrain, a study among the general population found that awareness of CRC symptoms and risk factors was low, with a score of 59% and 53%, respectively, resulting in an overall CRC knowledge score of 56% [31]. A national study in Saudi Arabia also showed low awareness of CRC-related symptoms and risk factors, with an overall mean awareness score of 11.05/23 among 5720 participants [32].

The results showed that level of education and kinds of job (part-time employees) were associated with overall awareness of colorectal cancer among our patients. Patients with higher education, including high school, university, and postgraduate degrees, had a statistically significant higher mean score of awareness compared to patients graduating from primary school. This result is consistent with other studies that found that better CRC knowledge among participants is associated with higher education levels [33,34].

Regarding kinds of job, part-time employed patients had a statistically significant lower overall awareness compared to full-time employed patients, with *p*-value = 0.015. This result may be explained by the fact that patients with full-time jobs are likely to have higher education levels and therefore better awareness compared to part-time job patients with lower levels of education. Moreover, older patients are more likely to have higher awareness of CRC compared to younger patients [31]. Patients with full-time jobs are usually older and have a higher risk of CRC due to aging and stress from work, which may also affect their awareness. Conversely, part-time employed patients are usually younger and may not be concerned about CRC as they are healthy and have not experienced CRC themselves or in their social circles.

## 5. Limitations

Our study has certain limitations. Given that it was conducted in a tertiary care hospital, the participants’ enrollment may not be fully representative of the entire population. The use of a cross-sectional study design raised concerns about missing data, especially regarding lifestyle changes and adherence. Additionally, due to unmeasured factors, we cannot entirely eliminate the influence of remaining confounding variables. To gain a more comprehensive understanding and evaluation of awareness related to the signs, symptoms, and risk factors of colorectal cancer among outpatients, a larger population study would be necessary.

## 6. Conclusions

This study highlights low awareness of CRC in the general population of Vietnam, particularly regarding modifiable risk factors. Efforts are needed to educate and raise awareness, especially among less educated populations. Further research is essential to enhance awareness of early CRC screening programs and reduce late-stage cases. This study underscores the need for health policy and public awareness initiatives to reduce the burden of this disease. To address this, health education programs and CRC-focused health promotion campaigns are recommended for the Vietnamese population.

## Figures and Tables

**Table 1 healthcare-11-03063-t001:** Sample demographic characteristics and history of cancer (N = 441).

Characteristic	N	%
Age group		
18–46	214	48.5
47–88	227	51.5
Gender		
Male	188	42.6
Female	253	57.4
Ethnicity		
Kinh ethnic	437	99.1
Other	4	0.9
Marital status		
Single	68	15.4
Married	365	82.8
Other	8	1.8
Education		
Elementary	45	10.2
Secondary school	77	17.5
High school	155	35.1
University	113	25.6
Postgraduate	19	4.3
Other	32	7.3
Job		
Employed full-time	148	33.6
Employed part-time	66	15.0
Unemployed	14	3.2
Self-employed	86	19.5
Full-time homemaker	82	18.6
Retired	35	7.9
Still studying	10	2.2
History of cancer		
Respondent’s cancer		
Yes	23	5.2
No	418	94.8
Partner’s cancer		
Yes	4	0.9
No	437	99.1
Close family’s cancer		
Yes	53	12.0
No	388	88.0
Relatives’ cancer		
Yes	18	4.1
No	423	95.9
Friend’s cancer		
Yes	17	3.8
No	424	96.2

**Table 2 healthcare-11-03063-t002:** Recall of CRC symptoms and risk factors.

CRC Symptoms	Recall (%)	CRC Risk Factors	Recall (%)
Change in bowel habits	36.7%	Pollution	66.9%
Abdominal pain	35.4%	Unhealthy/poor diet	53.7%
Weight loss	19.0%	Genes	50.6%
Back passage bleeding	18.1%	Lifestyle	41.7%
Feeling full	7.3%	Drinking alcohol	39.7%
Blood in stools	8.6%	Smoking	19.0%
Loss of appetite	6.6%	Not doing enough exercise	13.8%
Feeling bloated	6.3%	Stress	11.8%
Tiredness/anemia	3.6%	Not eating enough fruits and vegetables	8.8%
Lump	1.8%	Diet high in fat	3.4%
Change in stools color	1.4%	Eating red or processed meat	3.2%
Abdominal swelling	1.1%	Family history/relatives with cancer	2.7%
Don’t know	42.6%	Don’t know	6.1%

**Table 3 healthcare-11-03063-t003:** The mean score of awareness about CRC among outpatients (N = 441).

Variable	Mean	SD
Mean score of awareness about CRC signs and symptoms	3.46	2.91
Mean score of awareness about CRC risk factors	5.90	1.62
Overall mean score of awareness about CRC	9.36	4.53

**Table 4 healthcare-11-03063-t004:** The relationship between outpatients’ characteristics and their awareness about signs, symptoms, and risk factors of colorectal cancer (N = 441).

Characteristics	N	Signs and Symptoms			Risk Factors		
		Mean Score (95% CI)	t/F	*p*-Value	Mean Score (95% CI)	t/F	*p*-Value
Gender							
Male	188	3.34 (2.93 to 3.75)	−0.79	0.432	5.82 (5.60 to 6.04)	−0.93	0.355
Female	253	3.56 (3.19 to 3.93)			5.96 (5.76 to 6.17)		
Age group							
18–46	214	3.26 (2.86 to 3.66)	−1.44	0.151	6.00 (5.79 to 6.22)	0.20	0.201
47 and over	227	3.66 (3.28 to 4.04)			5.81 (5.59 to 6.03)		
Ethnicity							
Kinh	437	3.49 (3.21 to 3.76)	1.53	0.127	5.90 (5.75 to 6.06)	0.19	0.851
Other	4	1.25 (2.73 to 5.23)			5.75 (3.03 to 8.47)		
History of cancer							
Respondent’s cancer							
Yes	23	4.04 (2.67 to 5.42)	0.97	0.331	5.78 (4.91 to 6.66)	−0.36	0.717
No	418	3.44 (3.16 to 3.71)			5.91 (5.75 to 6.06)		
Partner’s cancer							
Yes	4	5.00 (2.40 to 7.60)	1.06	0.291	5.50 (2.19 to 8.81)	−0.50	0.62
No	437	3.45 (3.18 to 3.73)			5.91 (5.75 to 6.06)		
Close family’s cancer							
Yes	53	4.09 (3.24 to 4.95)	1.67	0.095	5.98 (5.57 to 6.39)	0.37	0.708
No	388	3.38 (3.09 to 3.67)			5.89 (5.73 to 6.06)		
Relatives’ cancer							
Yes	18	3.17 (1.57 to 4.76)	−0.45	0.656	5.94 (5.32 to 6.57)	0.11	0.911
No	423	3.48 (3.20 to 3.76)			5.90 (5.74 to 6.06)		
Friend’s cancer							
Yes	17	5.18 (4.00 to 6.36)	2.48	0.013 *	5.47 (4.74 to 6.20)	−1.12	0.265
No	424	3.40 (3.12 to 3.68)			5.92 (5.76 to 6.08)		
Marital status							
(1) Single	68	3.60 (2.79 to 4.42)	0.09	0.915	6.15 (5.70 to 6.59)	2.41	0.091
(2) Married	365	3.44 (3.15 to 3.73)			5.88 (5.72 to 6.04)		
(3) Other	8	3.50 (0.34 to 6.66)			4.88 (3.74 to 6.01)		
Level of education							
(1) Elementary	45	2.47 (1.61 to 3.33)	3.79	0.002 **	5.31 (4.86 to 5.76)	13.53	<0.001 ***
(2) Secondary school	77	3.01 (2.37 to 3.65)			5.14 (4.72 to 5.56)		
(3) High school	155	3.33 (2.90 to 3.75)			5.77 (5.55 to 5.99)		
(4) University	113	4.29 (3.71 to 4.88)	(4) > (1) ^a^		6.73 (6.47 to 6.98)	(4) > (1), (2), (3) ^a^;	
(5) Postgraduate	19	4.32 (2.82 to 5.81)			6.84 (6.19 to 7.49)	(5) > (1), (2) ^a^	
(6) Other	32	3.22 (2.20 to 4.24)			5.75 (5.06 to 6.44)		
Job							
(1) Employed full-time	148	3.51 (3.01 to 4.02)	2.65	0.015 *	6.24 (5.98 to 6.51)	3.35	0.003 **
(2) Employed part-time	66	2.83 (2.11 to 3.56)			5.55 (5.24 to 5.85)		
(3) Unemployed	14	3.14 (1.47 to 4.82)			4.93 (4.13 to 5.73)		
(4) Self-employed	86	3.00 (2.46 to 3.54)			5.93 (5.60 to 6.26)		
(5) Full-time homemaker	82	4.21 (3.56 to 4.85)			5.65 (5.26 to 6.03)		
(6) Retired	35	4.34 (3.45 to 5.23)			6.23 (5.60 to 6.86)		
(7) Still studying	10	2.30 (0.51 to 4.09)			5.30 (3.99 to 6.61)		

Note: * *p*-value < 0.05; ** *p*-value < 0.01; *** *p*-value < 0.001; ^a^ Scheffé’s method.

**Table 5 healthcare-11-03063-t005:** The associated factors of overall awareness of colorectal cancer among outpatients (linear regression) (N = 441).

Characteristics	Univariate Linear Analysis		
	Mean (SD)	Unadjusted Difference in Mean (95% CI)	*p*-Value
Gender			
Male	9.16 (3.62)	Reference	
Female	9.53 (4.00)	0.37 (−0.36 to 1.09)	0.323
Age group			
18–46	9.27 (3.79)	Reference	
47 and over	9.47 (3.91)	0.20 (−0.52 to 0.92)	0.585
Ethnicity			
Kinh	9.39 (3.85)	Reference	
Other	7.00 (3.16)	−2.39 (−6.19 to 1.4)	0.216
Marital status			
Single	9.75 (4.33)	Reference	
Married	9.32 (3.74)	−0.43 (−1.43 to 0.57)	0.399
Other	8.38 (4.37)	−1.38 (−4.20 to 1.45)	0.340
Level of education			
Elementary	7.78 (3.51)	Reference	
Secondary school	8.16 (3.89)	0.38 (−0.98 to 1.74)	0.584
High school	9.10 (3.35)	1.32 (0.09 to 2.54)	0.035 *
University	11.02 (3.94)	3.24 (1.96 to 4.52)	<0.001 ***
Postgraduate	11.16 (3.72)	3.38 (1.40 to 5.36)	0.001 **
Other	8.97 (4.00)	1.19 (−0.48 to 2.86)	0.163
Job			
Employed full-time	9.76 (4.18)	Reference	
Employed part-time	8.38 (3.38)	−1.38 (−2.48 to −0.27)	0.015 *
Unemployed	8.07 (3.75)	−1.69 (−3.78 to 0.41)	0.114
Self-employed	8.93 (3.36)	−0.83 (−1.84 to 0.19)	0.110
Full-time homemaker	9.85 (3.92)	0.10 (−0.93 to 1.13)	0.853
Retired	10.57 (3.74)	0.81 (−0.59 to 2.22)	0.255
Still studying	7.60 (3.34)	−2.16 (−4.6 to 0.29)	0.083
History of cancer			
Respondent’s cancer			
Yes	9.83 (4.71)	Reference	
No	9.34 (3.80)	−0.48 (−2.10 to 1.14)	0.559
Partner’s cancer			
Yes	10.5 (1.73)	Reference	
No	9.36 (3.86)	−1.14 (−4.94 to 2.66)	0.555
Close family’s cancer			
Yes	10.08 (3.99)	Reference	
No	9.27 (3.82)	−0.80 (−1.91 to 0.30)	0.155
Relatives’ cancer			
Yes	9.11 (4.04)	Reference	
No	9.38 (3.84)	0.27 (−1.55 to 2.09)	0.771
Friend’s cancer			
Yes	10.65 (2.91)	Reference	
No	9.32 (3.87)	−1.33 (−3.20 to 0.54)	0.163

Note: * *p*-value < 0.05; ** *p*-value < 0.01; *** *p*-value < 0.001.

## Data Availability

Data are contained within the article.

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
