# Peer review of "Assessing Awareness of Colorectal Cancer Symptoms among Outpatients: A Cross-Sectional Study at a Hospital in Vietnam"

_healthcare, 2023, doi:10.3390/healthcare11233063_

Round 1
Reviewer 1 Report
Comments and Suggestions for Authors - Overall, this is a good research study, however, there is a big need for significant grammar and style edits. It also needs a statement provided by the statistician that the analysis is accurate based on the variables selected.Comments on the Quality of English Language Overall, this is a good research study, however, there is a big need for significant grammar and style edits. It also needs a statement provided by the statistician that the analysis is accurate based on the variables selected.
Author Response
Responses:
- Thanks for your suggestions. We have made significant grammar and style edits to our paper based on reviewer comments and have indicated these changes with red words.
- Thanks for your suggestions We have conducted a thorough reevaluation of the statistical analysis in this study, ensuring its accuracy with respect to the selected variables. Additionally, Professor Chao-Hsien Lee, the corresponding author (https://orcid.org/0000-0001-9316-5098), served as the statistician and has also reviewed the statistical analysis, affirming its accuracy in relation to the chosen variables.
Reviewer 2 Report
Comments and Suggestions for Authors
The introduction should be more complete.
https://pubmed.ncbi.nlm.nih.gov/27268615/ and https://pubmed.ncbi.nlm.nih.gov/28283123/ may be useful to improve the introduction.
In method:
The method of sampling, the inclusion and exclusion criteria should be more complete.
convenience sample has many flaws, including lack of generalizability (To be considered in the discussion and limitations.)
Why 441?
In Data Collection Tools section: Patients or participants?
face-to-face interviews or Written questionnaire??
The validity and reliability of the questionnaire should be reported.
Study design 4 Present key elements of study design early in the paper Setting 5 Describe the setting, locations, and relevant dates, including periods of recruitment, exposure, follow-up, and data collection Participants 6 (a) Give the eligibility criteria, and the sources and methods of selection of participants Variables 7 Clearly define all outcomes, exposures, predictors, potential confounders, and effect modifiers. Give diagnostic criteria, if applicable Data sources/ measurement 8* For each variable of interest, give sources of data and details of methods of assessment (measurement). Describe comparability of assessment methods if there is more than one group Bias 9 Describe any efforts to address potential sources of bias Study size 10 Explain how the study size was arrived at Quantitative variables 11 Explain how quantitative variables were handled in the analyses. If applicable, describe which groupings were chosen and why (a) Describe all statistical methods, including those used to control for confounding (b) Describe any methods used to examine subgroups and interactions (c) Explain how missing data were addressed (d) If applicable, describe analytical methods taking account of sampling strategy Statistical methods 12 (e) Describe any sensitivity analyses
In table 1: what is Respondent’s cancer??
What is the difference Close Family’s cancer and Relatives’ cancer?
In table 2: 36.7% reported Change in bowel habits, it’s strange, how it's checkedØŸ
How you checked CRC Risk Factors? For example pollution?
In table 3 , what’s mean 9.36? Does that mean the average score is good?
Grading in the method should be clearly stated.
In title table 4 what is outpatients‘?
In the results, first the explanation should be presented, then the table should be presented.
Conclusions is long.
Informed Consent Statement: is missed.
After correcting the text, revise the abstract.
Author Response
Responses:
Thanks for your suggestions. We made the necessary changes as follows.
- We have thoroughly revised the manuscript and addressed your questions and comments one by one, as outlined below:
- Introduction: We have revised the Introduction according to your suggestion with useful articles you gave as references. We added second paragraph into Introduction section to make it more complete:
“Colorectal cancer represents a significant global health concern, with its incidence and mortality rates exhibiting notable variations across different regions. In the worldwide context, the research analyzed data from various countries and revealed that countries with higher Human Development Index scores, such as the Republic of Korea, Slovakia, and Hungary, had elevated colorectal cancer incidence rates [4]. This trend was similarly observed in the Asian-focused study, suggesting a consistency in the relationship between socio-economic development and colorectal cancer incidence [5]. These findings emphasize the importance of socio-economic factors, including Human Development Index, in understanding the prevalence of colorectal cancer, both worldwide and in the Asian region [4,5].”
References:
- Rafiemanesh H, Mohammadian-Hafshejani A, Ghoncheh M, Sepehri Z, Shamlou R, Salehiniya H, Towhidi F, Makhsosi BR. Incidence and Mortality of Colorectal Cancer and Relationships with the Human Development Index across the World. Asian Pac J Cancer Prev. 2016;17(5):2465-73. PMID: 27268615.
- Ghoncheh M, Mohammadian M, Mohammadian-Hafshejani A, Salehiniya H. The Incidence and Mortality of Colorectal Cancer and Its Relationship With the Human Development Index in Asia. Ann Glob Health. 2016 Sep-Oct;82(5):726-737. doi: 10.1016/j.aogh.2016.10.004. PMID: 28283123.
- Study design, Method of sampling: We have revised the method of sampling according to your suggestion:
“A cross-sectional study was conducted, involving a convenience sample of 441 participants who attended the Department of Day Service Unit at Hoa Hao Medic Co., Ltd between August and November 2021. Inclusion criteria for this study encompassed all Vietnamese individuals who availed themselves of the Department of Day Service Unit's services for general health check-ups during the data collection period and willingly agreed to participate. Exclusion criteria for the study were patients with a prior history of colorectal cancer or those facing mental and communication issues that impeded their ability to respond to information collection forms.
To mitigate potential biases inherent to cross-sectional studies, all participants meeting the inclusion criteria underwent face-to-face interviews using the Bowel CAM questionnaire. These interviews were conducted by trained interviewers during the participants' presence in the Day Service Unit.”
- We also added the Limitations which is the last paragraph in the Discussion section according to your suggestion:
Limitations
Our study has certain limitations. Given that it was conducted in a tertiary care hospital; the participants' enrollment may not be fully representative of the entire population. The use of a cross-sectional study design raised concerns about missing data, especially regarding lifestyle changes and adherence. Additionally, due to unmeasured factors, we cannot entirely eliminate the influence of remaining confounding variables. To gain a more comprehensive understanding and evaluation of awareness related to the signs, symptoms, and risk factors of colorectal cancer among outpatients, a larger population study would be necessary.
- Sample size is caculated based on the formula:
= 379
Where:
n: sample size.
Z2*(1-α/2) = 1.96 Corresponding with 95% reliability.
p = 0.56 (estimation of the percentage of people in society with estimation of the percentage of people in society with the rate of is 56% based on the overall knowledge score about CRC Nasaif and Al Quallaf (2018).
q = 1-p = 1 - 0.56 = 0.44
d: Absolute precision desired. Choosing d = 0.05.
Applying to the formula the sample size can calculate n = 379.
Take more 20% in case of missing data, then the total sample was 455. After cleaning data, only 441 samples were taken into the data analysis because of not replying all questionnaires.
- In Data Collection Tools section: We have corrected the word “Participants” instead of “Patients”
- Face-to-face interviews or Written questionnaire?
It’s face-to-face interview. We clearly stated it in the “Data collection tools” section
- The validity and reliability of the questionnaire: We have added the validity and reliability of the questionnaire in the “Data collection tools” section as following:
“This study utilized the Bowel CAM toolkit, version 2.1, developed by University College London and Cancer Research UK in 2011 and the validity was assessed by 16 bowel cancer specialists and 35 university experts. The toolkit demonstrated good internal reliability, with Cronbach's alpha exceeding 0.7 for all components (1).”
Reference:
- Cancer Research UK. Bowel Cancer Awareness Measure Toolkit Version 2.1 2011. https://www.cancerresearchuk.org/sites/default/files/health_professional_bowel_cam_toolkit_version_2.1_09.02.11.pdf (accessed on 24 March 2021).
- In table 1: "Respondent's cancer" implies that participants had a history of cancers other than colorectal cancer.
- The diffeence between Close Family’s cancer and Relatives’ cancer
- The term "close family" refers to a person's immediate family members, which typically includes their parents, children, and siblings.
- The term "relatives" encompasses a range of family connections, including parents-in-law, sons-in-law, daughters-in-law, and other individuals who share a family relationship through marriage or other familial connections.
- In table 2: 36.7% reported Change in bowel habits, it’s strange, how it's checked?
The question concerning "change in bowel habits" is a closed-ended question: “Do you think a change in bowel habits (diarrhea, constipation or both) over a period of weeks could be a sign of bowel cancer?” Respondents can choose from three options: "Yes," "No," or "Don't know." This response is obtained through self-report from participants via face-to-face interview. We provided participants with a clear explanation of what constitutes a change in bowel habits as per the question.
Here is a descriptive result of “Change in bowel habits” from SPSS analysis:
|
Change in bowel habits |
|||||
|
|
Frequency |
Percent |
Valid Percent |
Cumulative Percent |
|
|
Valid |
No |
279 |
63,3 |
63,3 |
63,3 |
|
Yes |
162 |
36,7 |
36,7 |
100,0 |
|
|
Total |
441 |
100,0 |
100,0 |
|
|
- How you checked CRC Risk Factors? For example pollution?
- In the questionnaire, CRC risk factors is an open question. The question was asked: What things do you think affect a person’s chance of developing bowel cancer? Prompt with ‘anything else?’ until the respondent cannot think of any more signs. If the person says they do not know any, prompt with ‘are you sure?’ and if necessary ‘take a minute to think about it’. Write down all of the risk factors that the person mentions exactly as they say it.
- For pollution: Participants answered that they believed various types of pollution, including air, water, and food, have the potential to contribute to the development of bowel cancer.
- In table 3, what’s mean 9.36? Does that mean the average score is good?
- The overall mean score for awareness about colorectal cancer (CRC) was determined by summing the mean scores for awareness of CRC signs and symptoms (question two) and the mean scores for awareness of CRC risk factors (question six). These scores were computed exclusively for participants' correct responses to the respective closed-ended (prompted) questions, specifically question two and question six. The total scores ranged from zero to 19, with a maximum score of 9 for signs and symptoms and 10 for risk factors. In this study, the overall mean awareness score for CRC was found to be 9.36/19.
- Therefore, mean the average score is 36/19 which is low (not good). This result highlighted the consistent trend of low awareness regarding CRC symptoms and risk factors in the region, underscoring the need for targeted awareness campaigns and education initiatives.
- Grading in the method should be clearly stated.
We have added grading in the method section under “Data collection tools” as following: “ The total scores, calculated from a combination of 19 items, ranged from zero to 19. It consisted of 9 items for signs and symptoms and 10 items for risk factors, scored as 1 point for a "correct" response, 0 points for an "incorrect" response, and 0 points for an "unclear" response.”
- Title table 4: Outpatients refer to individuals who visit the general clinics of Hoa Hao Medic Co., Ltd for general health check-ups and willingly agree to take part in the study.
- Results: We have revised the results section in the manuscript following your suggestion by presenting the explanations first, followed by the corresponding tables.
- Conclusions: We have revised the “Conclusion” to be shorter according to your suggestion.
- Informed Consent Statement: We have added the Informed Consent Statement: “All participants provided written informed consent to take part in this study”
- Abstract: We have revised the Abstract after correcting the text as following:
“Abstract: Colorectal cancer (CRC) is a prevalent cancer globally, including in Vietnam where its incidence is rapidly increasing. The aim of this study was to evaluate the awareness of signs, symptoms, and risk factors of colorectal cancer among outpatients at Hoa Hao Medic Company Limited, Ho Chi Minh City, Vietnam. A cross-sectional study was conducted, and a total of 441 people visiting Hoa Hao Medic Company Limited for a general health check-up and voluntarily agree to participate in this study were recruited through convenience sampling method. Data was collected through face-to-face structured interviews using the Bowel CAM questionnaire. As the results, the highest percentage of well-recalled warning signs was "change in bowel habit" (36.7%), followed by abdominal pain (35.4%). Other warning signs such as weight loss and rectal bleeding were reported by 19.0% and 18.1% of participants, respectively. Over 42% of participants stated that they were unaware of any signs or symptoms. The most commonly identified risk factors for CRC were pollution (66.9%), genetics (50.6%), and an unhealthy/poor diet (53.7%) for unprompted questions. The overall awareness score of participants was 3.46/9 (SD ±2.91) for signs and symptoms of CRC and 5.90/10 (SD ±1.62) for risk factors. Univariate linear regression identified education level and occupation as predictors of higher CRC awareness. In conclusion, the study highlights that overall awareness of CRC is low among outpatients at Hoa Hao Medic Company Limited. Strategies to increase awareness, knowledge, and education programs are necessary to promote early detection of CRC and reduce its burden in Vietnam.”
Reviewer 3 Report
Comments and Suggestions for Authors
I have the following comments:
1) Methods: The sampling frame of the outpatient clinic is not representative of the population as these patients are already linked with the healthcare system for some reason.
overall the manuscript is well written but doesn't add much to existing literature.
Author Response
Responses:
1.Thanks for your suggestions. We have made the necessary changes and incorporated the “Limitations” into this paper.
Limitations
Our study has certain limitations. Given that it was conducted in a tertiary care hospital; the participants' enrollment may not be fully representative of the entire population. The use of a cross-sectional study design raised concerns about missing data, especially regarding lifestyle changes and adherence. Additionally, due to unmeasured factors, we cannot entirely eliminate the influence of remaining confounding variables. To gain a more comprehensive understanding and evaluation of awareness related to the signs, symptoms, and risk factors of colorectal cancer among outpatients, a larger population study would be necessary.
Reviewer 4 Report
Comments and Suggestions for Authors
Evaluating public literacy regarding an increasingly important health problem, such as Colon Cancer, is relevant to base decisions about interventions aiming for earlier detection and reduction of burden and impact. Adapting strategies to regional specificities is also important
You present and justify your study clearly. Methodologies are adequate and statistical tools also. Results and discussion are fine. Conclusions aligned wit the findings
I wonder and would like to see an explanation why asking about symptoms, and it wasn't asked about what is the best way to detect colon cancer, nor the possible consequences of missing or delay the diagnosis. Risk Facr
Author Response
Responses:
Thanks for your affirmation and appreciation of our research. I would like to answer your question:
Vietnam is experiencing a significant rise in Colorectal Cancer (CRC) cases, recording over 14,000 new cases and approximately 7,000 CRC-related fatalities annually (1). Notably, many of these cases are diagnosed in advanced stages with distant metastases, significantly impacting survival rates. Research has demonstrated that early-stage CRC diagnosis can result in a five-year survival rate of up to 90% (2). Several prior studies have highlighted the critical role of disease awareness in influencing participation in screening and promoting timely diagnosis (3,4,5). This emphasizes the importance of improving public knowledge about primary symptoms and risk factors. Enhanced awareness can lead to increased engagement in clinical screening programs, ultimately alleviating the disease burden, reducing treatment costs, mitigating socioeconomic consequences, and minimizing productivity losses (6). Our current study concentrates solely on assessing awareness levels concerning CRC symptoms and risk factors among outpatients. In light of our research findings, this study provides valuable insights and recommendations for healthcare professionals and policymakers, with a focus on strategies such as disseminating information and implementing health education programs to enhance awareness of this preventable disease.
References:
- Vietnam National Cancer Hospital. Vietnam update of colorectal cancer diagnosis and treatment progress 2019. https://benhvienk.vn/cap-nhat-cac-tien-bo-ve-chan-doan-va-dieu-tri-ung-thu-dai-truc-trang-tai-viet-nam-nd85316.html (accessed 5 Jun 2022).
- Gulf Center for Cancer Control and Prevention, Executive Board of Health Ministers’ Council For The Gulf Cooperation Council State and The Research Cneter King Faisal Specialist Hospital and Research Center – Riyadh. Ten-Year Cancer Incidence Among Nationals of the GCC States 1998-2007. https://www.moh.gov.bh/Content/Files/Publications/GCC%20Cancer%20Incidence%202011.pdf (accessed 10 April 2023).
Round 2
Reviewer 2 Report
Comments and Suggestions for Authors
-
Reviewer 3 Report
Comments and Suggestions for Authors
I believe they added limitations to the discussion, so is ok from my side
Reviewer 4 Report
Comments and Suggestions for Authors
Thank you for considering the reviewer's comments. You have made an extensive revision that clarifies the document. The information obtained is useful for defining literacy-improving strategies